# ACIQ: Analytical Clipping for Integer Quantization of neural networks

## Abstract

Unlike traditional approaches that focus on the quantization at the network level, in this work we propose to minimize the quantization effect at the tensor level. We analyze the trade-off between quantization noise and clipping distortion in low precision networks. We identify the statistics of various tensors, and derive approximate analytical expressions for the mean-square-error degradation due to clipping. By optimizing these expressions, we show marked improvements over standard quantization schemes that normally avoid clipping. For example, just by choosing the accurate clipping values, more than 40% accuracy improvement is obtained for the quantization of VGG16-BN to 4-bits of precision. Our results have many applications for the quantization of neural networks at both training and inference time. One immediate application is for a rapid deployment of neural networks to low-precision accelerators without time-consuming fine tuning or the availability of the full datasets.

## 1 Introduction

A significant drawback of deep learning models is their computational costs. Low precision is one of the key techniques being actively studied recently to conquer the problem. With hardware support, low precision training and inference can compute more operations per second, reduce memory bandwidth and power consumption, and allow bigger network to fit into a device.

In general, a low-precision scheme involves a floating-point to integer conversion, which introduces quantization noise into the network. This quantization noise is strongly linked to the dynamic range, defined as the range between the largest and smallest values that need to quantized. For a given $N$-bit integer representation, a smaller dynamic range leads to a smaller spacing between the $2^N$ quantization levels, enabling improved resolution and smaller quantization noise. To reduce this quantization noise, the dynamic range can be limited by clipping the values in the tensor. This clipping process introduces an additional noise because of the loss of information that otherwise would be carried by the clipped portion of the tensor. Hence, a trade-off between clipping and quantization effects exist. To find the best clipping value we need to minimize the information loss.

In this paper, we study the effect of clipping with the aim of improving overall quantization noise. To this end, we first study the distribution of values within these tensors. In all our measurements, the statistical distributions of weights and activations are observed to follow a bell-curve. This indicates that large values occur very rarely compared to small values, and suggests that the loss of information due to the clipping process might be compensated by improving the resolution of the more common smaller values.

To optimize this process further, it is essential to understand the underlying distribution of tensor elements before applying the clipping. By running a few statistical tests, we were able to see on a variety of convolution models that activation tensors follow either a Gaussian or Laplacian distributions with a high degree of certainty (p-value $< 0.01$). This modeling of activation tensors enables a clear formulation of the quantization process and constitutes the first step for its optimization.

We turn to consider the objective we aim to optimize. It is well known that when batch norm is applied after a convolution layer, the output is invariant to the norm of the output on the proceeding layer [Ioffe & Szegedy (2015)] i.e., $BN(C \cdot W \cdot x) = BN(W \cdot x)$ for any given constant $C$. This quantity is often described geometrically as the norm of the activation tensor, and in the presence of

this invariance, the only measure that needs to be preserved upon quantization is the *directionality* of the tensor. Therefore, quantization preserves tensor information if the angle between the high-precision tensor and its quantized version is small. Recently, Banner et al. (2018) has shown that this angle depends only on the quantization error power (L2 - norm) and the power of original tensor. Therefore, minimizing the power of the quantization error constitutes a plausible goal for the optimization of the quantized network in terms of accuracy.

In Section 4, we provide a rigorous formulation to optimize the quantization effect of activation tensors using clipping by analyzing both the Gaussian and the Laplace priors. This formulation is henceforth refered to as Analytical Clipping for Integer Quantization (ACIQ).

These analytical results have many applications for the quantization of neural networks at both training and inference time. For example, a straightforward quantization of the weights and activations to 8-bit fixed point representation has been shown to have a negligible effect on the model accuracy. Yet, in the majority of the applications, further reduction of precision quickly degrades performance, calling for an optimal clipping scheme to minimize information-loss during quantization. On a more general level, exploiting the statistics of the activation tensors to minimize their quantization toll is orthogonal to other techniques for network quantization. It can work in synergy with other schemes to achieve more than could have been achieved by each individually. Finally, it is easy to implement and requires only the adjustment of clipping value according to an analytical formula.

We further demonstrate the applicability and usability of our analytic terms on the following challenging problem. Given a pre-trained network, we would like to quantize the weights to 8-bit of precision and most activations to 4-bits of precision without any further processing (e.g., re-training). This specific setting is of a particular interest due to quantization of activations to 4-bits, which alleviates a major bottleneck in terms of memory bandwidth. Prior attempts using standard techniques Krishnamoorthi (2018) show severe degradation on accuracy. While several recent works were able to overcome this issue by additional re-training McKinstry et al. (2018), this is not feasible in many practical settings, e.g., we often do not have the dataset on which the network is working on.

We compare ACIQ against two methods: (i) the traditional method that avoids clipping (also known by gemmlowp Jacob et al. (2017b)), where values are uniformly quantized between the largest and smallest tensor values; (ii) the iterative method suggested by NVIDIA to search for a good clipping threshold based on the Kullback-Leibler Divergence (KLD) measure Migacz (2017). Results are summarized in Table 1. While both ACIQ and gemmlowp are fast non-iterative methods, ACIQ significantly outperforms in terms of validation accuracy. On the hand, KLD is an exhaustive time-consuming procedure, which iteratively evaluates the KLD measure on a large candidate set of clipping values, and then returns the clipping value for which best evaluation is attained. In our simulations ACIQ and gemmlowp require a single pass over tensor values, while KLD requires 4000 passes. Nonetheless, excluding ResNet-101, ACIQ outperforms KLD in terms of validation accuracy.

The methods introduced in this work may be additionally useful to current and future applications, such as the attempts to fully train in a low precision setting Banner et al. (2018). In this scenario, intermediate activations and weights are known to have their statistics change throughout the training process, further highlighting the need for a precise and careful scale determination.

| Model | Dataset | Baseline FP32 | No-clipping | KLD | ACIQ |
|---|---|---|---|---|---|
| VGG16 | ImageNet | 71.59% | 53.9% | 67.04% | **67.4%** |
| VGG16-BN | ImageNet | 73.36% | 29.5% | 65.85% | **67.6%** |
| ResNet-18 | ImageNet | 69.75% | 53.2% | 65.06% | **65.8%** |
| ResNet-50 | ImageNet | 76.1% | 52.7% | 70.8% | **71.45%** |
| ResNet-101 | ImageNet | 77.3% | 50.8% | **71.7%** | 69.53% |
| Inception v3 | ImageNet | 77.2% | 41.4% | 59.25% | **60.8%** |

Table 1: Validation accuracy of different post-training quantization schemes (8-bit weights, 4-bit activations). ACIQ runs 4000 times faster compared to KLD and, excluding ResNet-101, outperforms KLD in terms of validation accuracy.

.

## 2 PREVIOUS WORK

In many cases, taking a model trained for full precision and directly quantizing it to 8-bit precision, without any re-training, can result in a relatively low loss of accuracy Jacob et al. (2017a); Gong et al. (2018); Krishnamoorthi (2018). More recently, Banner et al. (2018); Wu et al. (2018) has shown that 8-bit precision is sufficient not just for running trained models, but also for training them. Yet, naively quantizing a full precision model below 8-bits usually incurs significant accuracy degradation.

Many attempts have been made to diminish this effect, but they usually employ training of the model with quantization constraints or modifying network structure Lin et al. (2017). Recent work has been able to quantize weights and activations to 4-bits, but requires a long training time of 110 epochs McKinstry et al. (2018). Other concepts to enable quantization below 8-bits have been proposed Rastegari et al. (2016); Zhou et al. (2016); Choi et al. (2018). Per-channel scale has shown to be important both for inference Rastegari et al. (2016) and for training Wu et al. (2018). The concept of non-uniform quantization has recently been suggested by Park et al. (2018) and Baskin et al. (2018).

To the best of our knowledge, there have been only a few attempts to clip activations before. Nvidia proposed an iterative method to search for a good clipping threshold based on the Kullback-Leibler divergence measure Migacz (2017). More recently, Choi et al. (2018); Jung et al. (2018) have proposed an activation clipping parameter that is optimized during training. Finally, Wu et al. (2018) has proposed a heuristic approach to adjust the clipping values according to the desired precision. As contrast to previous works, we propose a simple and efficient method to find the optimal clipping threshold analytically from the distribution of the tensor, by minimizing MSE error. Our method takes into account not only the target precision but also the statistics of the tensor. We prove analytically that, for a given distribution, our method finds an optimal solution for threshold that minimizes the MSE error. We then demonstrate its usefulness on several Imagenet topologies.

## 3 PROBABILITY DISTRIBUTION FITTING FOR ACTIVATIONS

It has already been noted by prior arts that neural network distributions are near Gaussian in practice, sometimes further controlled by procedures such as batch normalization Soudry et al. (2014). In this section, we construct an estimate of the underlying probability density function of the activation tensors. Traditionally, no clipping was made in standard GEMMLWOP. Hence, quantization levels are uniformity spaced between the largest and the smallest values in the tensor. Yet, this approach is non-optimal, due to the fact that the activation tensors have bell-shaped distributions. Therefore, to reduce quantization noise (or increase resolution), it might be desirable to clip the values above a certain threshold. Specifically, we need to find the best clipping value that, on the one hand, maintains a low clipping rate, but, on the other hand, improves resolution. To do so we must first understand the underlying data distribution.

To that end, we collect the data of various tensors at different layers. We observe that the data is in general symmetrically distributed around a mean. We next estimate the goodness of fit to several bell-shaped distributions. This is done by measuring the static distance (largest vertical line) between the cumulative distribution function (CDF) of the empirically observed distribution and the CDF of the reference distribution (also know by Kolmogorov-Smirnov test Lopes (2011). By considering the activations of all layers of ResNet50 on ImageNet, we obtain the average static distance to each of the following distributions, with $p$-value$< 0.01$:

| Distribution | Static Distance |
|---|---|
| Laplace | 0.070 |
| Normal | 0.053 |
| Logistic | 0.150 |
| Cauchy | 0.142 |
| Uniform | 0.490 |
| Loglaplace | 0.505 |
| Lognorm | 0.540 |

As one can see, the best fit is established by the Laplace and Normal distributions. In figure 4 (see Appendix), we plot the normalized distributions of the activation tensors at different layers for Resnet50, and compare them against both priors.

## 4  AN ANALYTICAL MODEL OF TENSOR QUANTIZATION

In this section, we provide a detailed analysis for establishing the optimal clipping values under either Gaussian or Laplace distributions. We first derive a generic expression for any given distribution for the expected MSE as a function of clipping value. We then use this expression to develop a specific expression for each distribution. Finally, we establish the optimal clipping values by solving the equations for which the derivative with respect to the clipping value are set to zero.

Let $X$ be a high precision random variable with a probability density function $f(x)$. Without loss of generality, we assume a prepossessing step has been made so that the average value in the tensor zero i.e., $\overline{X} = \mu = 0$ (we do not lose generality since we can always subtract and add this mean). Assuming bit-width $M$, we would like to quantize the values in the tensors uniformally to $2^M$ discrete values.

Commonly (e.g., in GEMMLOWP), integer tensors are uniformly quantized in the range $[-\alpha, \alpha]$, where $\alpha$ is determined by the tensor maximal absolute value. In the following we show that the this choice of $\alpha$ is suboptimal, and suggest a model where the tensor values are clipped to reduce quantization noise. For any $x \in \mathbb{R}$, we define the clipping function $\text{clip}(x, \alpha)$ as follows

$$\text{clip}(x, \alpha) = \begin{cases} x & \text{if } |x| \le \alpha \\ \text{sign}(x) \cdot \alpha & \text{if } |x| > \alpha \end{cases} \tag{1}$$

Denoting by $\alpha$ the clipping value, the range $[\alpha, -\alpha]$ is partitioned to $2^M$ equal quantization regions. Hence, the quantization step $\Delta$ between two adjacent quantized values is established as follows:

$$\Delta = \frac{2\alpha}{2^M} \tag{2}$$

Our model assumes values are rounded to the midpoint of the region (bin) i.e., for every index $i \in [0, 2^M - 1]$ all values that fall in $[-\alpha + i \cdot \Delta, -\alpha + (i+1) \cdot \Delta]$ are rounded to the midpoint $q_i = -\alpha + (2i + 1)\frac{\Delta}{2}$, as illustrated in Figure 1. Then, the expected mean-square-error between $X$ and its quantized version $Q(X)$ can be written as follows:

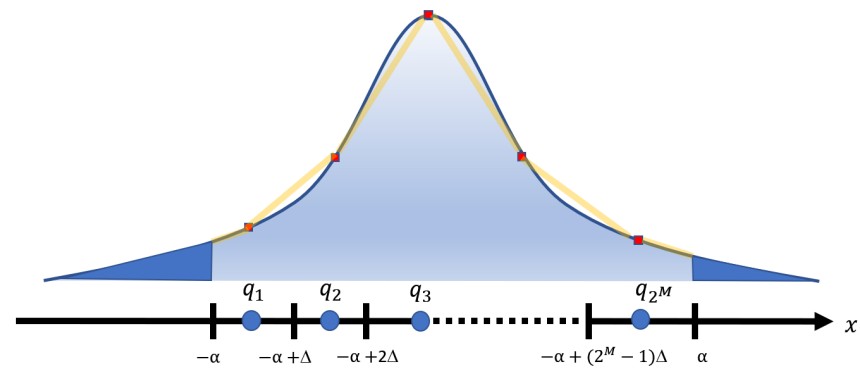

Figure 1: uniform quantization

$$E[(X - Q(X))^2] =$$

$$= \int_{-\infty}^{-\alpha} f(x) \cdot (x + \alpha)^2 dx +$$

$$+ \sum_{i=0}^{2^M - 1} \int_{-\alpha + i \cdot \Delta}^{-\alpha + (i+1) \cdot \Delta} f(x) \cdot (x - q_i)^2 dx + \quad (3)$$

$$+ \int_{\alpha}^{\infty} f(x) \cdot (x - \alpha)^2 dx$$

Equation 3 is composed of three parts. The first and last terms quantify the contribution of $\text{clip}(x, \alpha)$ to the expected mean-square-error. Note that for symmetrical distributions around zero (e.g., Gaussian $N(0, \sigma^2)$ or Laplace$(0, b)$) these two terms are equal and their sum can therefore be evaluated by multiplying any of the terms by 2.

The second term corresponds to the expected mean-square-error when the range $[-\alpha, \alpha]$ is quantized uniformly to $2^M$ discrete levels. We approximate the density function $f$ by a construction of a piecewise linear function whose segment breakpoints are points in $f$, as illustrated in Figure 1. Since we consider only smooth probability density functions (e.g., Gaussian or Laplace), the resulting approximation error is small for sufficient resolution i.e., small quantization step size $\Delta$. In the appendix we show that given a density function $f$, the quantization noise can be approximated as follows:

$$\sum_{i=0}^{2^M - 1} \int_{-\alpha + i \cdot \Delta}^{-\alpha + (i+1) \cdot \Delta} f(x) \cdot (x - q_i)^2 dx \approx \frac{2 \cdot \alpha^3}{3 \cdot 2^{3M}} \cdot \sum_{i=0}^{2^M - 1} f(q_i) \quad (4)$$

Equation 4 represents the rounding error (as opposed to clipping error) due to the rounding of all values in the bin $i$ to its center $q_i$. For sufficient resolution and a smooth density function, we found that the uniform distribution in the range $[-\alpha, \alpha]$ enables much simpler analysis with little effect on the accuracy. In subsection 4.3, we show that with this assumption the analytic results are in a good agreement with the simulation results. Similar observations have already been noted in previous works (e.g., Marco & Neuhoff (2005)). By substituting the uniform density function $f(x) = \frac{1}{2\alpha}$ into Equation 4, the following simpler rounding error can be computed:

$$\sum_{i=0}^{2^M - 1} \int_{-\alpha + i \cdot \Delta}^{-\alpha + (i+1) \cdot \Delta} f(x) \cdot (x - q_i)^2 dx \approx \frac{2 \cdot \alpha^3}{3 \cdot 2^{3M}} \cdot \sum_{i=0}^{2^M - 1} \frac{1}{2\alpha} = \frac{\alpha^2}{3 \cdot 2^{2M}} \quad (5)$$

By substituting Equation 5 into Equation 3, and using the symmetrical argument mentioned above, Equation 3 can be simplified for symmetrical distributions as follows:

$$E[(X - Q(X))^2] = \frac{\alpha^2}{3 \cdot 2^{2M}} + 2 \cdot \int_{\alpha}^{\infty} f(x) \cdot (x - \alpha)^2 dx \quad (6)$$

In the following we provide a closed form solution for the case where the density probability distribution function $f(x)$ is either Gaussian $N(0, \sigma^2)$ or Laplace$(0, b)$.

## 4.1 LAPLACE CLIPPING

In the following we develop an expression based on Equation 6 for the Laplace case. Since we assume that $\mu = 0$, we have the following Laplace density function $f(x) = \frac{1}{2b} e^{-\frac{|x|}{b}}$. In order to derive a closed form solution for Equation 6, we need to evaluate

$$2 \cdot \int_{\alpha}^{\infty} f(x) \cdot (x - \alpha)^2 dx. \quad (7)$$

Let $\Psi(x)$ represent the expression below:

$$\Psi(x) = -\frac{1}{2} \left[ x^2 + 2(b - \alpha) x + 2b^2 - 2\alpha b + \alpha^2 \right] e^{-\frac{x}{b}} \quad (8)$$

By taking the derivative of $\Psi(x)$ with respect to $x$, it is easy to see that $\Psi(x)$ is the correct antiderivative of the integrand in equation 7. Hence,

$$\int_{\alpha}^{\infty} f(x) \cdot (x - \alpha)^2 dx = \Psi(\inf) - \Psi(\alpha) = b^2 \cdot e^{-\frac{\alpha}{b}}$$

We can finally state Equation 6 for the laplace case as follows.

$$E[(X - Q(X))^2] \approx 2 \cdot b^2 \cdot e^{\frac{-\alpha}{b}} + \frac{2 \cdot \alpha^3}{3} \cdot \sum_{i=0}^{2^M - 1} f(q_i) = 2 \cdot b^2 \cdot e^{\frac{-\alpha}{b}} + \frac{\alpha^2}{3 \cdot 2^{2M}} \tag{9}$$

Finally, in order to find the $\alpha$ that give the minimum MSE, the corresponding derivative with respect to $\alpha$ is set equal to zero as follows:

$$\frac{\partial E[(X - Q(X))^2]}{\partial \alpha} = \frac{2\alpha}{3 \cdot 2^{2M}} - 2be^{-\frac{\alpha}{b}} = 0 \tag{10}$$

## 4.2 GAUSSIAN CLIPPING

We now turn to evaluate Equation 6 for the Gaussian case. Given a Gaussian random variable $X \sim N(0, \sigma^2)$, we define $\Psi(x)$ to represent the expression below:

$$\Psi(x) = \frac{\left(\alpha^2 + \sigma^2\right) \operatorname{erf}\left(\frac{x}{\sqrt{2}\sigma}\right)}{2} - \frac{(x\sigma - 2\alpha\sigma) e^{-\frac{x^2}{2\sigma^2}}}{\sqrt{2\pi}} \tag{11}$$

As in subsection 4.1, one can observe that by taking the derivative of $\Psi(x)$ with respect to $x$, it is easy to show that $\Psi(x)$ is the correct antiderivative of Equation 7 for the case where $f$ represents the Gaussian density function i.e., $f(x) = \frac{1}{\sqrt{2\pi}\sigma} e^{-\frac{x^2}{2\sigma^2}}$. Next, we use $\Psi(x)$ on the range $[\alpha, \infty]$ to evaluate Equation 7 for the Gaussian case as follows:

$$\int_{\alpha}^{\infty} f(x) \cdot (x - \alpha)^2 dx = \Psi(\infty) - \Psi(\alpha)$$

$$= \frac{\alpha^2 + \sigma^2}{2} \cdot \left[1 - \operatorname{erf}\left(\frac{\alpha}{\sqrt{2}\sigma}\right)\right] - \frac{\alpha \cdot \sigma \cdot e^{-\frac{\alpha^2}{2 \cdot \sigma^2}}}{\sqrt{2\pi}}$$

Equation 6 can thus be written for the case of Gaussian distribution as follows:

$$E[(X - Q(X))^2] \approx \frac{\alpha^2 + \sigma^2}{2} \cdot \left[1 - \operatorname{erf}\left(\frac{\alpha}{\sqrt{2}\sigma}\right)\right] - \frac{\alpha \cdot \sigma \cdot e^{-\frac{\alpha^2}{2 \cdot \sigma^2}}}{\sqrt{2\pi}} + \frac{\alpha^2}{3 \cdot 2^{2M}} \tag{12}$$

In order to find the optimal clipping values for which mean-square-error is minimized, we need to differentiate $E[(X - Q(X))^2]$ with respect to $\alpha$ and set the derivative equal to zero as follows.

$$\frac{\partial E[(X - Q(X))^2]}{\partial \alpha} = \alpha \left[1 - \operatorname{erf}\left(\frac{\alpha}{\sqrt{2}\sigma}\right)\right] - \frac{\sigma^2 e^{-\frac{\alpha^2}{2\sigma^2}}}{\sqrt{2\pi}\sigma} - \frac{\sigma e^{-\frac{\alpha^2}{2\sigma^2}}}{\sqrt{2\pi}} + \frac{2\alpha}{3 \cdot 2^{2M}} \tag{13}$$

## 4.3 NUMERICAL EVALUATIONS

In figure 2 we introduce the mean-square-error as a function of clipping value for various bit widths. We present the theoretical derivations expressed by Equations 9 and 12, and compare them against the MSE resulting from the clipping and quantization of 10,000 values, generated from a Gaussian/Laplace distribution. Note that theoretical results are in a good agreement with the experiments. As expected, the difference occurs only for very low-bit width and large clipping values where the uniform assumption tends to break. Code to replicate these experiments appears online here: https://github.com/submission2019/AnalyticalScaleForIntegerQuantization

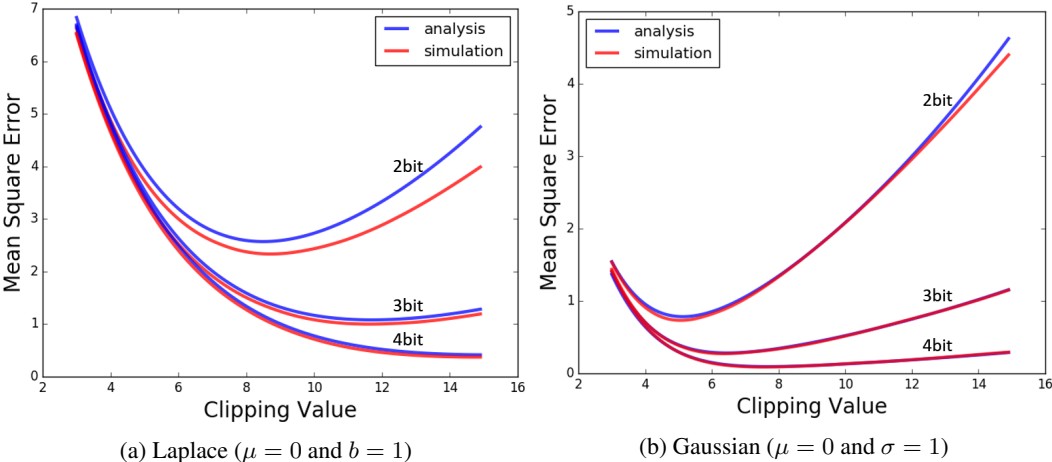

(a) Laplace ($\mu = 0$ and $b = 1$)   (b) Gaussian ($\mu = 0$ and $\sigma = 1$)

Figure 2: Expected mean-square-error as a function of clipping value for different quantization levels. Good agreement between theory and simulations.

## 5 EXPERIMENTS

We evaluate the proposed ideas of analytical clipping on multiple models, and demonstrate its usefulness by suggesting a fast method to quantize neural network aggressively and without a significant accuracy drop. Our method is simple and efficient, and does not require re-training. During quantization we only adjust the clipping values according to our analytical formula, and compare it against the standard GEMMLOWP quantization Jacob et al. (2017b) that avoids clipping. We consider a mixed precision scheme of 4 and 8 bits quantization. All weights are kept at 8-bit precision. The goal is to quantize as many activation layers as possible to 4-bits of precision without a significant accuracy degradation. This setting is important as activation tensors constitute a major memory bottleneck for inference in large models. The code to replicate all our experiments is available online.[1]

We use equations 9 and 12 to evaluate the expected mean-square-error (MSE). Based on the expected MSE and a quantization requirement (i.e., the number of activation layers that need to be quantized to 4-bits), we decide whether to keep the activation tensor at 8-bits of precision or quantize it into a 4-bit representation. These activation tensors are clipped before the ReLU. We keep all weight tensors at 8-bit precision. Finally, to facilitate calculations, we avoid calculating Equations 9 and 12 from scratch each time. Rather, as $\mathcal{N}(0, \sigma^2) = \sigma \cdot \mathcal{N}(0, 1)$ and $Laplace(0, b) = b \cdot Laplace(0, 1)$, it is sufficient to store the optimal clipping values for $\mathcal{N}(0, 1)$ and $Laplace(0, 1)$ and scale these values by $\sigma$ and $b$, which are estimated from the tensor values.

We ran experiments to see how many activation layers can be quantized from 8 to 4 bits of precision without a significant accuracy drop. These experiments have been made with ResNet-18, ResNet-50, ResNet-101, VGG-16, VGG-16 with batch normalization and Inception-v3 on the ImageNet dataset. For each method, we select $N$ layers (tensors) to be quantized to 4 bits using our optimized clipping method and compare it against the standard GEMMLOWP approach. In figure 3 we present this accuracy-quantization tradeoff. Our results clearly show the significant potential of our optimized clipping method. For example, just by choosing the correct clipping values, more than half of the layers can be quatized to 4 bits with an accuracy drop of less than 1% for Res18, Res50 and both versions of Vgg16. Also, unlike the standard approach, degradation is much more predictable and follow a roughly linear trend. Therefore, by adjusting the number of layers, we can conveniently and linearly control the accuracy-quantization trade-off, as can be seen by the linear relation shown in Figures 3g, 3h and 3i.

---

[1] https://github.com/submission2019/AnalyticalScaleForIntegerQuantization

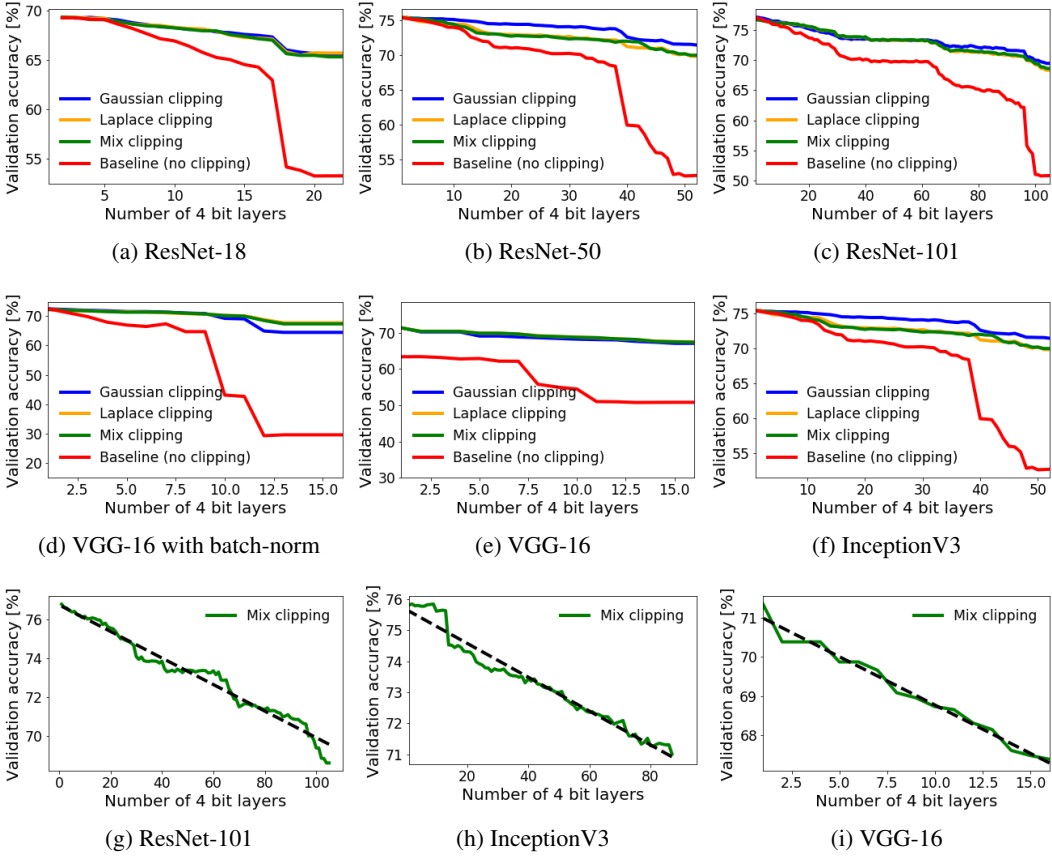

Figure 3: Accuracy degradation as a function of quantization level: much smaller and predictable accuracy degradation with optimal clipping. Red curve represents the standard GEMMLOWP quantization method; yellow curve represents the accuracy of our method, assuming Laplace distribution; blue curve represents the accuracy of our method, assuming Gaussian distribution; green curve represents the accuracy of our method when using both types of distributions; the distribution with the better estimate for the mean-square-error is selected (predicted either by Equation 9 or Equation 12).

## 6 CONCLUSION

We introduce ACIQ - an optimized clipping framework for improved quantization of neural networks. Optimized clipping is shown to have a drastic impact on quantization in a variety of models. The underlying reason lies in the statistical dispersion of activations, where large values occur very rarely. We show the bell-curve statistics of activations are best fit as either Laplace or Gaussian distributions, and formulate the clipping process as an optimization problem. The solution to this optimization problem constitutes a polynomial-exponential equation that can be calculated numerically for a variety of statistical parameters, and stored in a lookup table for fast retrieval. This scheme is very simple and easy to implement either in software or in hardware.

While results are very encouraging, this work is only the first step on the ladder for successful deployment of clipping in neural networks. First, our main focus in this work is quantization of activations, while similar evaluation still needs to be done for weights. On a more general level, our framework is not restricted to the inference settings and can be extended to training. For example, our preliminary results show that quantization of gradients might benefit from the clipping of small values (i.e., sparsification). Establishing the correct threshold for gradients is yet another important direction for future work. While much work still needs to be done with regards to optimized clipping, we believe our work clearly demonstrates the major importance of this concept for the quantization of neural networks.

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

# Appendices

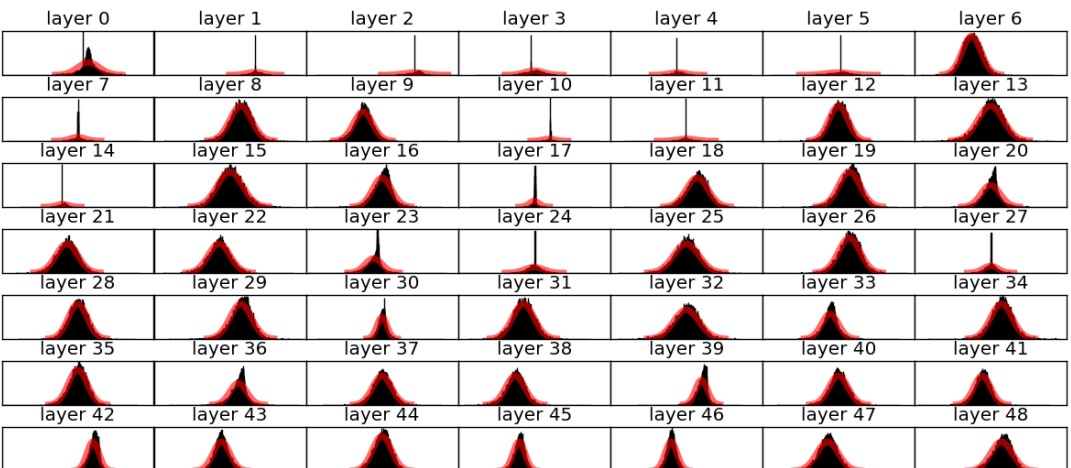

(a) Fitting the activations with a Gaussian Distribution

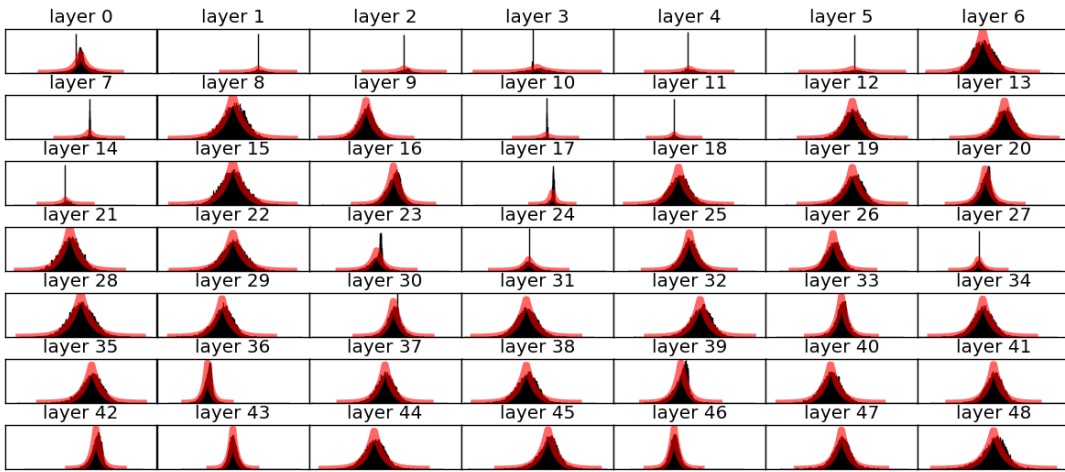

(b) Fitting the activations with a Laplace Distribution

Figure 4: Fitting activation tensors to Gauss and Laplace distribution at different ResNet50 layers. The statistical fit to both models is acceptable and similar.

## A    PIECE-WISE LINEAR APPROXIMATION

Here we provide a more accurate analysis related to the qunatization noise (i.e., the second term in Equation 3), measured as the expected mean-square-error when the range $[-\alpha, \alpha]$ is quantized uniformly to $2^M$ discrete levels. To that end, we approximate the density function $f$ by a construction of a piece-wise linear function $g$ such that $f(q_i) = g(q_i)$ for each $i \in [0, 2^M - 1]$. Since we consider only smooth probability density functions (e.g., Gaussian or Laplace), the resulting approximation error is small for sufficient resolution i.e., small quantization step size $\Delta$. In figure 1 we provide an illustration for this construction.

We turn to calculate the linear equation for each line segment of the piece-wise linear function $g$, falling in the range $[-\alpha + i \cdot \Delta, -\alpha + (i+1) \cdot \Delta]$. To that end, we consider the slope (derivative) and the value of the density function at the midpoint $q_i$. With these two values we can define for each segment $i \in [0, 2^M - 1]$ the corresponding form of linear approximation:

$$g(x) = f(q_i) + \frac{df}{dx}(q_i) \cdot (x - q_i), \qquad -\alpha + i \cdot \Delta \leq x \leq -\alpha + (i+1) \cdot \Delta \tag{14}$$

We now turn to calculate the second term in Equation 3. By equation 14, and since $q_i$ is defined to be the midpoint between the integration limits, the following holds true

$$
\begin{aligned}
\sum_{i=0}^{2^M-1} \int_{-\alpha+i\cdot\Delta}^{-\alpha+(i+1)\cdot\Delta} f(x) \cdot (x-q_i)^2 dx &\approx \sum_{i=0}^{2^M-1} \int_{-\alpha+i\cdot\Delta}^{-\alpha+(i+1)\cdot\Delta} g(x) \cdot (x-q_i)^2 dx = \\
&= \sum_{i=0}^{2^M-1} \int_{-\alpha+i\cdot\Delta}^{-\alpha+(i+1)\cdot\Delta} f(q_i) \cdot (x-q_i)^2 + \sum_{i=0}^{2^M-1} \int_{-\alpha+i\cdot\Delta}^{-\alpha+(i+1)\cdot\Delta} \frac{df}{dx}(q_i) \cdot (x-q_i)^3 dx = \\
&= \frac{1}{3} \sum_{i=0}^{2^M-1} f(q_i) \cdot (x-q_i)^3 \Big|_{-\alpha+i\cdot\Delta}^{-\alpha+(i+1)\cdot\Delta} + \frac{1}{4} \sum_{i=0}^{2^M-1} \frac{df}{dx}(q_i) \cdot (x-q_i)^4 \Big|_{-\alpha+i\cdot\Delta}^{-\alpha+(i+1)\cdot\Delta} = \\
&= \frac{\Delta^3}{12} \sum_{i=0}^{2^M-1} f(q_i) = \frac{2 \cdot \alpha^3}{3 \cdot 2^{3M}} \cdot \sum_{i=0}^{2^M-1} f(q_i)
\end{aligned}
\tag{15}
$$

