# OpenReview forum: "ACIQ: Analytical Clipping for Integer Quantization of neural networks"
_ICLR.cc/2019/Conference_

### Official Review · AnonReviewer2 · 2018-10-29
**A simple but not very convincing clipping method for activation quantization in deep networks**

**Rating:** 5
**Confidence:** 4

**Review:**

This paper empirically finds that the distribution of activations in quantized networks follow  Gaussian or Laplacian distribution, and proposes to determine the optimal clipping factor by minimizing the quantization error based on the distribution assumption.

The pros of the work are its simplicity, the proposed clipping and quantization does not need additional re-training. However, while the key of this paper is to determine a good clipping factor, the authors use uniform density function to represent the middle part of both Gaussian and Laplacian distributions where the majority of data points lie in, but exact computation for the tails of the distributions at both ends. Thus the computation of quantization error is not quite convincing. Moreover, the authors do not compare with the other recent works that also clip the activations, thus it is hard to validate the efficacy of the proposed method.

For the experiments, the authors mention that a look-up table can be pre-computed for fast retrieval of clipping factors given the mean and sigma of a distribution.  However, the mean and sigma are continuous numbers, how is the look-up table made?  Moreover, how is the mean and std estimated for each weight tensor and what is  the complexity?

---

> ### Author Response · Authors · 2018-11-26
> **Response to reviewer 2**
>
> Reviewer  addressed three concerns:
>
> 1. It has been observed by many prior arts that quantization error can be assumed to be uniformly distributed (e.g., http://daniel-marco.com/Academic%20Files/Additive%20Noise%20Model.pdf). Our expression estimates the MSE as a combination of the MSE that results from the quantization error in the “middle part” (i.e., uniformly distributed) and the MSE that results from the clipping error at the tails of the distribution (i.e., laplacian/gaussian). We verify this assumptions using synthetic simulations that clearly show that this type of approximation is accurate in practice (see figure 2). We also provide the code for this simple synthetic experiment here: https://github.com/submission2019/AnalyticalScaleForIntegerQuantization . Finally, we now have a more general expression that estimates the MSE of any density function at the middle part using a piece wise linear approximation, enabling to use not only uniform distributions at the middle part.
>
> 2. Since submission, we made a comparison against the only previous method we are aware of: the Kullback-Leibler Divergence (KLD) clipping method suggested by NVIDIA (see update for table 1). Our approach runs 4000 times faster compared to KLD and, excluding ResNet-101, outperforms KLD in terms of validation accuracy.
>
> 3. We agree that the mean and sigma are continuous numbers. But the correct clipping value can be calculated by scaling the optimal clipping value for the standard gaussian distribution N(0,1) by sigma. We use this trick in our simulations. We improved the explanation of this issue (see Section 5  - end of the second paragraph)

---

### Official Review · AnonReviewer1 · 2018-10-31

**Rating:** 4
**Confidence:** 4

**Review:**

This paper derives a formula for finding the minimum and maximum clipping values for uniform quantization which minimize the square error resulting from quantization, for either a Laplace or Gaussian distribution over pre-quantized value. This seems like too small a contribution to warrant a paper. I wasn't convinced that appropriate baselines were used in experiments. There were a number of statements that I believed to be technically slightly incorrect. There were also some small language problems (though these didn't hinder understanding).

more specific comments:

abstract:
"derive exact expressions" -- these expressions aren't exact. they turn out to be based on a piecewise zeroth order Taylor approximation to the density.

main paper:
"allow fit bigger networks into" -> "allow bigger network to fit into"
"that we are need" -> "that need"
"introduces an additional" -> "introduces additional"
clippig -> clipping

it's not clear a-priori that information loss is the property to minimize that maximizes performance of the quantized network.

"distributions of tensors" -> "distribution of tensor elements"
this comment also applies in a number of other places, where the writing refers to the marginal distribution of values taken on by entries in a tensor as the distribution over the tensor. note that a distribution over tensors is a joint distribution over all entries in a tensor. e.g. it would capture things like eigenvalues, entry-entry covariance, rather than just marginal statistics.

"than they could have by working individually" -> "than could have been achieved by each individually"

Why the focus on small activation bit depth? I would imagine weight bit-depth was more important than activation bit depth. Especially since you're using ?32-bit? precision in the weight/activations multiplications, so activations are computed at a high bit depth anyways.

Table 1: Give absolute accuracies too! Improvement relative to what baseline?

sec 2:
sufficeint -> sufficient
\citep often used when it should instead be \citet.
"As contrast" -> "In contrast"

section 3:
uniformity -> uniformly

I don't believe the notion of p-value is being used correctly here w.r.t. the Kolmogorov-Smirnov test.

Figure 1: The mean square error should never go to 0. This suggests something is wrong. If it's just a scaling issue, consider a semilogy plot.

Figure 2: I'm unclear what baseline (no clipping) refers to in terms of clipping values. For uniform quantization there needs to be some min and max value.

---

> ### Author Response · Authors · 2018-11-26
> **Response to reviewer 1**
>
> The paper indeed provides a formula for optimal quantization when the distribution of tensor elements is either laplace or gauss. The paper also shows the relevance of these derivations to a very attractive use-case i.e., the conversion of full precision network to low precision network without time-consuming re-training or the availability of the full datasets. Our approach is shown to have significant advantages over previous approaches (as summarized in Table 1).
>
> Response to more specific comments:
> Language problems: We have incorporated all typos and paraphrasing suggestions
>
>
> 1.“it's not clear a-priori that information loss is the property to minimize that maximizes performance of the quantized network.”  The connection between quantization error and classification accuracy has been investigated through the preservation of the direction of the quantized tensor. See for example here:
>          a.	https://arxiv.org/pdf/1805.11046.pdf#page=9&zoom=100,0,96 (section 5.1)
>          b.	https://arxiv.org/pdf/1705.07199.pdf (section 3.1)
> We have added a detailed explanation about the connection between power of the quantization error and accuracy drop (see paragraph #5 in the introduction).
>
> 2.“Give absolute accuracies too! Improvement relative to what baseline?” We now provide the baselines we use in our experiments (see Table 1).
>
> 3. “The mean square error should never go to 0. This suggests something is wrong. If it's just a scaling issue, consider a semilogy plot.”: This was indeed a scale issue only. It is not relevant anymore (the figure was removed and replaced by the synthetic experiments showing that analysis and simulations are in a good agreement).
>
> 4.	“I'm unclear what baseline (no clipping) refers to in terms of clipping values. For uniform quantization there needs to be some min and max”: we improved the explanation of this issue in the introduction (see beginning of paragraph 9), where we explain that the traditional method that avoids clipping uniformly quantize the values between the largest and smallest tensor elements.

---

### Official Review · AnonReviewer3 · 2018-11-12
**Errors and Contributions not significant**

**Rating:** 4
**Confidence:** 5

**Review:**

The paper describes a clipping method to improve the performance of one particular type of quantization method that is naive clipping to closest "bins". The contribution of the paper is the (possibly incorrect) derivation of the clipping value that causes the least quantization error IF assumptions can be made about the distribution of the parameters (in a non-bayesian sense). Thus, the significance is low due to both reasons.

One conceptual issue is the assumed relationship between quantization error and classification accuracy. The literature has shown that high quantization error does not necessarily mean low classification accuracy when using non-uniform quantization. The proposed clipping does not account for classification accuracy (on training set), but I understand the motivation being that the training set is not available.

1. There seems to be an error in derivation of Eq (3), the first term should be $(x-sgn(x).\alpha) = x+\alpha$ for $x$ negative. Please comment on this.

2. When solving the integrals, the authors simply pull the solution "out of the hat" and show that the derivative is the integrand. This is a very opaque presentation that we cannot see how you solved the integral. What is C in $\psi(x)$?

3. The assumptions on the parameters are only valid for the particular model/dataset/precision. The assumption does not generalize arbitrarily. For example, models with quantized weights have bi-modal distributions. How would you clip the  activations after e.g. a ReLu? This is without going in to the weaknesses of the K-S test.

4. Experiments do not show any comparison to the large body of prior work in this area.

5. Page 4, para below (3), what is "common additive orthogonal noise"? You should explain or give intuition instead of simply referring to a different paper.

6. In the uniform case, one would think f(x)=1/<range of the interval>=2\alpha. Why is it 1/\Delta?

6. Section 4, range should be [-\alpha, \alpha] instead of [\alpha, -\alpha]? Since \alpha is positive.

---

> ### Author Response · Authors · 2018-11-26
> **Response to reviewer 3**
>
> We conduct synthetic experiments showing analysis is in a very good agreement with synthetic simulations when distributions of tensor elements are either Laplace or Normal (see figure 2 in our new submission). The code to replicate these sanity checks appears here: https://github.com/submission2019/AnalyticalScaleForIntegerQuantization. We also improved the presentation of section 4 and provide a new figure to make the analysis easier to understand.
>
> As noted by many prior arts, neural network distributions, are near Gaussian in practice, sometimes further controlled by procedures such as batch normalization. See for example here:
> 1. https://arxiv.org/pdf/1804.10969.pdf
> 2. https://openreview.net/pdf?id=B1IDRdeCW
> 3. https://papers.nips.cc/paper/5269-expectation-backpropagation-parameter-free-training-of-multilayer-neural-networks-with-continuous-or-discrete-weights.
> In addition, we were able to see these bell-shaped distributions through both statistical tests (KS-test at section 3) and the visual appearance of the histograms (see appendix).
>
> The connection between quantization error and classification accuracy has been investigated through the preservation of the direction of the quantized tensor. See for example here:
> 1. https://arxiv.org/pdf/1805.11046.pdf#page=9&zoom=100,0,96 (section 5.1)
> 2. https://arxiv.org/pdf/1705.07199.pdf (section 3.1)
> We have added a detailed explanation about the connection between power of the quantization error and accuracy drop (see paragraph #5 in the introduction).
>
> Detailed comments:
> 1. Typo corrected.
> 2. For correctness, we believe it is enough to provide the primitive functions of these integrals (since this can be verified by differentiation). The direct derivations are long unnecessary tedious calculations. Also, C in $\psi(x)$ is the standard constant of integration for indefinite integrals. To avoid unclarity we removed this constant now from the text.
> 3. We disagree with this comment. We have validated our work on six different models and improvement was dramatic with respect to gemmlowp for the quantization of activation tensors.  Weight tensors are kept at 8 bit of precision so the bi-modal distribution of weights does not apply to our work. The activations are clipped before the ReLU as we clarify now in Section 5 (second paragraph).
> 4. Since submission, we made a comparison against the only previous method we are aware of: the Kullback-Leibler Divergence (KLD) clipping method suggested by NVIDIA (see update for table 1). Our approach runs 4000 times faster compared to KLD and, excluding ResNet-101, outperforms KLD in terms of validation accuracy.
> 5. We provide now a better intuition for the analysis in section 4.
> 6. For the uniform case, f(x) = 1/(2*alpha). We explicitly mention that in the paper now (just before equation 5).
> 7. Typo corrected.

---

> > ### Comment · AnonReviewer3 · 2018-11-30
> > **reply to response**
> >
> > 1. Re the distribution assumption, the response from the authors is not convincing. The paper you mentioned (https://arxiv.org/pdf/1805.11046.pdf#page=9&zoom=100,0,96) says that, when using BN, "quantization preserves the direction (angle) of high-dimensional vectors when W follows a Gaussian distribution", this has nothing to do with your assumption that W follows a gaussian distribution.
> >
> > The original question was not that "gaussian -> low quantization error -> good performance" (I think this is clear in the past 3 years) but rather "non-gaussian -> high quantization error -> bad performance?". Recent work suggests this may not always lead to bad performance (e.g. there are binary models with good performance and high quantization error).
> >
> > What does Figure 5 show? That quantization error is similar for analysis and simulation. Is this level of error "small"? Clearly, it depends on the number of bits. The gaussian assumption is not true for lower bit networks (the paper you referred uses 8 bits). Overall, the distribution assumption is a weakness.
> >
> > 3. The point was about more datasets like VOC, beyond image classification.
> >
> > Thank you for improving the paper, I have increased my rating appropriately.

---

> > > ### Author Response · Authors · 2018-12-01
> > > **Reply to remaining issues**
> > >
> > > The paper we mention (i.e.,  https://arxiv.org/pdf/1805.11046.pdf#page=9&zoom=100,0,96 ) assumes Gaussian distribution and construct a solution that couldn’t work unless tensors have approximately a Gaussian distribution. Due to the central limit theorem, neural network distributions are not general. In practice, tensors have a bell shape distribution where small values are much more frequent compared to the large values. Recent efforts take this prior into account and design quantizers with improved accuracy (e.g., https://arxiv.org/pdf/1804.10969.pdf). Our clipping method uses this bell-shaped distribution (we focus on Gauss/Laplacian distributions) to give higher precision where we need it (i.e., small values) at the expense of truncating very few large values. At the bottom line, our simulations prove that much better accuracy can be obtained with these assumptions. In all six evaluated models, we gain at least 12% validation accuracy improvement (in VGG16-BN we get 38% improvement) compared to GEMLOWP, which doesn’t assume anything about the data distribution and quantize according to max()-min().
> > >
> > > Figure 5 does not appear in the paper. From the context, we guess the reviewer refers to Figure 2. The figure shows that analysis is in a good agreement with simulations, and for each bit-width there exists a distinct minimum at a certain clipping value.  The reviewer makes the following statment: “The gaussian assumption is not true for lower bit networks (the paper you referred uses 8 bits)” Here is a paper that takes the Gaussian assumption for binary networks to explain why binary networks work in terms of high dimensional geometry (see page 2 about angle preservation property of random tensors from Gaussian distributions): https://arxiv.org/pdf/1705.07199.pdf

---

### Public Comment · ~Evgenii_Zheltonozhskii1 · 2018-09-28
**Explanation and comparison of resutls**

Hi, can you explain the numbers you present in Table 1? What do you compare to? Also, I haven't manage to find comparison to any other quantization paper, neither numbers of accuracy you achieved for any network. I've run your code and acquired 65.75% top-1 and 86.70% top-5 for ResNet-18. However, recent work, such as PACT (https://arxiv.org/abs/1805.06085) and LQ-nets (https://arxiv.org/abs/1807.10029) achieve significantly higher results for much coarser quantization - 69+% top-1 for 4 bit for both activation and weights.

P.S. The caption of subfigure 2f is missing.

---

> ### Author Response · Authors · 2018-09-28
> **reply to explanation and comparison of results**
>
> We are not suggesting a quantization approach at the network level. Rather, we try minimizing the quantization effect at the tensor level only. We claim that when tensor values exceed a certain threshold, they should be clipped. Our main result is an analytical formula for clipping these values that, depending on the statistical distribution of the tensor, finds the *optimal* threshold (with respect to mean-square-error).
>
> We focus on clipping only (i.e., no re-training or fine-tuning). Hence, we compare against the standard integer quantization approach that avoids clipping (i.e., GEMMLOWP). This serves as the baseline for the comparison in Table 1.  You mention 65.75% top-1 accuracy for Res18 using our analytical clipping. Without clipping you would have 53.2% top-1 accuracy. We also have a similar result for the VGG-16 model, where we show that by just clipping correctly the activation tensors, you could gain more than 40% accuracy improvement compared to the case of no clipping.
>
> Finally, we are fully aware of the recent works you mention (in fact, PACT paper is cited and explained). Yet, these are completely not in the settings of our work. Both papers *learn* a good quantization through training.  Our work is orthogonal and can work in synergy with these techniques. You can minimize the effect of quantization at the tensor level and, at the same time, compensate for quantization at the network level using training/fine-tuning. There are many other applications. For example, a rapid deployment of neural networks trained in full precision to low precision accelerators without having the full datasets on which the networks are working on.
>
> The missing caption of subfigure 2f refers to Inception_v3.

---

> > ### Public Comment · ~Evgenii_Zheltonozhskii1 · 2018-09-29
> > **Thanks**
> >
> > Thanks for clarification!

---

### Meta-Review · Area_Chair1 · 2018-12-17
**incremental work**

**Confidence:** 5
**Recommendation:** Reject

**Metareview:**

The paper describes a clipping method to improve the performance of quantization. The reviewers have a consensus on rejection due to the contribution is not significant.